# Computational Fluid Dynamics Prediction of External Thermal Loads on Film-Cooled Gas Turbine Vanes: A Validation of Reynolds-Averaged Navier–Stokes Transition Models and Scale-Resolving Simulations for the VKI LS-94 Test Case

Simone Sandrin [1], Lorenzo Mazzei [1,*], Riccardo Da Soghe [1] and Fabrizio Fontaneto [2]

[1] Ergon Research s.r.l., via Campani 50, 50127 Florence, Italy; simone.sandrin@ergonresearch.it (S.S.); riccardo.dasoghe@ergonresearch.it (R.D.S.)

[2] Department of Turbomachinery and Propulsion, von Karman Institute for Fluid Dynamics, 1640 Rhode Saint Genèse, Belgium; fabrizio.fontaneto@vki.ac.be

\* Correspondence: lorenzo.mazzei@ergonresearch.it

**Abstract:** Given the increasing role of computational fluid dynamics (CFD) simulations in the aerothermal design of gas turbine vanes and blades, their rigorous validation is becoming more and more important. This article exploits an experimental database obtained by the von Karman Institute (VKI) for Fluid Dynamics for the LS-94 test case. This represents a film-cooled transonic turbine vane, investigated in a five-vane linear cascade configuration under engine-like conditions in terms of the Reynolds number and Mach number. The experimental characterization included inlet freestream turbulence measured with hot-wire anemometry, aerodynamic performance assessed with a three-hole pressure probe in the downstream section, and vane convective heat transfer coefficient distribution determined with thin-film thermometers. The test matrix included cases without any film-cooling injection, pressure-side injection, and suction-side injection. The CFD simulations were carried out in Ansys Fluent, considering the impact of mesh sizing and steady-state Reynolds-Averaged Navier-Stokes (RANS) transition modelling, as well as more accurate transient scale-resolving simulations. This work provides insight into the advantages and drawbacks of such approaches for gas turbine hot-gas path designers.

**Keywords:** turbomachinery; gas turbines; aerodynamics; heat transfer; film cooling; CFD; stress-blended eddy simulation; turbulence

## 1. Introduction

The field of gas turbine design has undergone a significant transformation in recent years, driven largely by advancements in computational fluid dynamics (CFD). The evolution of computational power has not only expanded the scope of CFD but also revolutionized its application in the design and optimization of gas turbines. The advent of enhanced computational capabilities has enabled three-dimensional simulations of industrially relevant test cases, and the (unsteady) Reynolds-Averaged Navier–Stokes (RANS) approach has become the standard modeling technique for most gas turbine components [1]. Moreover, the advancement of gas turbine design has placed a growing emphasis on comprehending and predicting with high fidelity the lifespan of the components, particularly those in the hot-gas path. Therefore, the role of CFD has transcended beyond traditional aerodynamic considerations to encompass the critical aspect of wall heat transfer, which is vital for the longevity and efficiency of gas turbine components. This aspect is particularly crucial in the context of high-pressure turbine vanes/blades, where temperature gradients at the walls significantly influence the overall performance and lifespan of the turbine [2]. However, accurately predicting wall heat transfer in the aggressive environments of gas turbines, characterized by the presence of technological devices such as cooling holes and

tip gaps, remains a tough challenge. Additionally, turbulence plays a major role in heat transfer. Turbulence modeling is one of the main sources of uncertainty in CFD simulations of technical flows. This is not surprising, as turbulence is one of the most complex aspects of classical physics. Turbulence is generated in turbines by many phenomena: flow perturbations from the combustor, shear interaction, secondary flows, coolant and purge flow injection, stator/rotor interaction, and boundary layer transition.

Laminar-to-turbulent boundary layer transition is indeed a very complex phenomenon that can significantly affect the aero and thermal loads on gas turbine airfoils and can be classified depending on the cause that triggers the transition. The most relevant types of transition in gas turbines are as follows:

- Bypass Transition: This occurs when the laminar BL is disrupted by significant external disturbances, such as high levels of freestream turbulence, abruptly accelerating the transition. For example, in turbomachinery blading, attached boundary layers are exposed to relevant freestream turbulence levels, and the transition is likely to be triggered by Klebanoff streaks.
- Oblique Transition: This is a 3D phenomenon typically initiated by the leading edge or surface imperfections. These disturbances then grow moving downstream and form oblique wave patterns, eventually breaking down into turbulent flow.
- Separated-Flow Transition: An adverse pressure gradient leads to the separation of the laminar BL from the surface, generating a free shear layer. Its instability causes the transition and the reattachment in the form of a turbulent BL. This transition type is relevant for turbine blades with high-lift characteristics, operating at high angles of attack, or in off-design conditions.
- Shock-Induced Transition: This is particularly relevant in high-speed gas turbines, where the transition is triggered by shock waves interacting with the boundary layer. The shock wave causes a sudden increase in boundary layer thickness and eventually the transition.

Each of these transition types has different implications for the design and operation of gas turbines. At the same time, the inherent complexity makes them difficult to reproduce with high fidelity in CFD simulations, especially in the RANS approach, in which the turbulent characteristics of the flow are modeled through a turbulence model. Therefore, RANS-based models consist of a crude representation of these physical phenomena. Over the last decade, significant progress has been made in the field of RANS transition modeling, which features algebraic models as well as models with additional transport equations. Among the many models available in the literature, the two most used transition models are probably the $k - k_l - \omega$ by Walters and Cokljat [3] and the $\gamma - Re_\theta$ by Langtry and Menter [4]. Leaving aside the details of the physico-mathematical formulation, the first model uses an additional transport equation for the laminar kinetic energy (LKE), which enables taking into account the pre-transitional rise of the fluctuating kinetic energy. The second model is instead a two-equation transition model used to modify turbulent transport equations to simulate laminar, laminar-to-turbulent, and turbulence states in a fluid flow. The $\gamma - Re_\theta$ model does not aim to model the physics of the problem but attempts to fit a wide range of experiments and transition methods into its formulation. The transition model calculates an intermittency factor that creates (or extinguishes) turbulence by slowly introducing turbulent production at the laminar-to-turbulent transition location, as described by Langtry [5]. An evolution of the $\gamma - Re_\theta$ model is the $\gamma$ model, also known as the Intermittency model, developed by Menter et al. [6]. In the $\gamma$ transition model formulation, the transport equations are reduced from two to one. All the empirical correlations that enter the model are now computed using only local quantities.

Transition models have been widely used to estimate the heat transfer distribution on gas turbine vanes and blades affected by the laminar-to-turbulent transition.

In a study that compared the heat transfer predictions on the C3X vane, it was found that the $k - k_l - \omega$ model and the $\gamma - Re_\theta$ model did not accurately predict the transition onset and remained at laminar heat transfer levels at a lower freestream turbulence intensity

of 0.5%. When the freestream turbulence intensity was increased to 20%, the $\gamma - Re_\theta$ model made accurate predictions and captured both the transition onset and length. Unfortunately, the $k - k_l - \omega$ model has not been as successful at this level [7]. The $\gamma - Re_\theta$ model has been highly successful in capturing the onset of transition in various studies [8–10]. According to Papa et al. [8], this model has shown excellent heat transfer predictions on the suction surface in three-dimensional flow regimes near endwalls. However, the authors also reported that the model tends to overpredict heat transfer at the leading edge, which may be due to excessive turbulence production at the stagnation point. In another study, the $\gamma - Re_\theta$ transition model was evaluated using a database of multiple low-pressure turbine blades. The model was able to accurately predict the flow around blades with mild diffusion rates, considering factors such as losses, outflow angle, and wall shear stress. However, it underpredicted the results for blades with high diffusion rates and significant separation. For more information on diffusion rates, please refer to the original source [11]. A recent study [12] aimed to analyze the heat transfer and transition predictions of the $\gamma - Re_\theta$ model in a low-pressure turbine stage. The results suggested that the model could predict the spanwise distribution of transition onset. However, the $\gamma - Re_\theta$ model predicted a late transition in the hub region and a somewhat early transition at the midspan, although it generally captured the overall trends. Over time, the $\gamma - Re_\theta$ model has proven to be successful, but the numerical results have not shown a perfect match with experimental data. This is not surprising, given the correlation-based nature of this transition model. The $k - k_l - \omega$ model has not received as much attention in the literature for heat transfer-related applications as the $\gamma - Re_\theta$ transition model, making its applicability for predicting the heat transfer coefficient (HTC) on a turbine vane difficult. An analysis was conducted with the LKE-based model in a vane cascade, but the results were not compared with those of the $\gamma - Re_\theta$ model. According to the authors, the model was able to accurately predict high Reynolds number flow, but its agreement with experimental data for low Reynolds number flow was limited [13]. Another study by Liu [14] focused on the transition performance for low Reynolds number flow applications, where the $\gamma - Re_\theta$ model was found to provide better transition onset predictions. A study was conducted on a NACA airfoil with a laminar separation bubble that re-attaches as turbulent [15]. The study examined the effectiveness of the $\gamma - Re_\theta$, $\gamma$, and $k - k_l - \omega$ models. Among these three models, only the $\gamma - Re_\theta$ model accurately captured the laminar separation bubble, although the reattachment point was slightly underpredicted. The gamma model performed similarly to the original SST turbulence model, and the inclusion of the transition model did not provide any benefit for separation-induced transition. The study concluded that the $k - k_l - \omega$ model required the highest computational effort and was therefore rejected. However, it was also noted that the model does not have any explicit sensor for separation-induced transition, unlike for bypass and natural transition. A sensor for separation-induced transition in the form of the wall-distance Reynolds number ($Re_y$) was incorporated into the $k - k_l - \omega - I$ model in [10]. Dick and Kubacki concluded in their review that all transition models have strengths and weaknesses [10]. The authors evaluated the $\gamma - Re_\theta$ model and concluded that it is generally a good choice but needs improvement for separation-induced transition, in contrast with the above findings. The $k - k_l - \omega$ model was said to perform equally as well as the $\gamma - Re_\theta$ model, except for separation-induced transition, where the $\gamma - Re_\theta$ model produced too much turbulence.

Pacciani et al. [16] extensively used the $k - k_l - \omega$ model to study separated flow in the high-lift, low-Reynolds-number cascades T106C and T108, tested at VKI. Additional sensitivity analyses were provided to assess the effects of the Reynolds number, expansion ratio, and freestream turbulence on isentropic Mach-number distributions and total pressure defect. Pacciani et al. [17] focused instead on the T120 cascade, characterized by the formation of a laminar separation bubble on the SS. Heat transfer measurements and predictions were compared covering different Mach and Reynolds numbers under both steady and periodic unsteady inflow conditions. A handful of articles have studied the Mark II blade from NASA, solely focusing on the $\gamma - Re_\theta$ model, revealing that the transi-

tion model can accurately characterize the laminar zone of the leading edge concerning the heat transfer coefficient. However, the predictive capability deteriorates moving toward the trailing edge after the transition [18–20].

Enico [21] conducted a study on the midspan heat transfer coefficient predictions of uncooled turbine guide vanes using RANS turbulence modeling. The study was based on the experimental data from the LS-89 von Karman Institute test case. The study revealed that the $\gamma - Re_\theta$ transition model underpredicted the transition onset as well as the heat transfer coefficient distribution on the suction side of the vane. Additionally, the model poorly predicted the pressure side HTC for turbulence intensities between 4 and 6%.

Despite their practical relevance, there is a lack of research on the impact of transition models on the calculation of the HTC in the context of film-cooled gas turbine vanes. Among the few works on this topic, it is worth citing Daugulis [22], who incorporated turbine vane endwalls into the model and investigated the HTC at different locations in the spanwise direction of the vane based on experiments conducted by Giel et al. [23]. The main difference in geometry was the incorporation of the turbine endwalls and more pronounced vane curvature. He concluded that transition modeling is needed to accurately capture the transition onset at low turbulence intensities of less than 1%. In such cases, the $\gamma - Re_\theta$ method was implemented. However, for turbulence intensities higher than 8%, transition modeling was not deemed necessary due to the early development of a fully turbulent boundary layer on the suction side. Furthermore, Johnsson and Asiegbu [24] studied the capabilities of the Shear Stress Transport (SST) $k - \omega$ turbulence model with the $\gamma - Re_\theta$ transition model to predict flow features, focusing on the VKI experimental test case of the film-cooled LS-94 vane. The model proved capable of estimating the heat transfer coefficient well, mainly on the suction side of the vane, whereas the pressure side was largely underpredicted.

This paper investigates the prediction capabilities of the three transition models in comparison with the fully turbulent $k - \omega$ SST RANS turbulence model for heat transfer prediction in a film-cooled highly loaded turbine guide vane. The test case is the LS-94 vane, specifically the film-cooled version of the well-documented airfoil from the VKI [25], representative of a modern turbine vane. This airfoil has been extensively used for CFD validation [26–28] and recently upgraded to provide experimental data in the presence of film cooling (see Fontaneto's works [29,30]). Given the increasing interest in Large-Eddy Simulations (LES) or LES-like approaches for turbine aerothermal applications, especially when the combustor is coupled with the nozzle guide vane. For this purpose, it was decided to include in the comparison a transient simulation carried out with the Stress-Blended Eddy Simulation (SBES) model, a hybrid RANS/LES approach implemented in Ansys Fluent.

The discussion around this specific objective is organized as follows. Section 2 details the main geometrical parameters of the test case and describes the experimental configuration. Details of the models, numerical setup, and computational domain are presented in Section 3, along with the outcomes of the mesh sensitivity analysis. The results are then compared with experimental data for the estimation of the heat transfer coefficient along the vane chord in Section 4. More specifically, comparisons are made to estimate the capability of the numerical methods to predict the heat transfer coefficient distribution when film cooling is present at the pressure side, at the suction side, or absent. Finally, Section 5 presents the conclusions.

## 2. Test Case

This section describes the geometry used for this study, outlining the specific dimensions and configuration of the experimental setup. It offers a clear understanding of the physical context in which the experiments are focused. Overall, the description aims to offer readers a comprehensive understanding of the vane design, allowing them to appreciate the relevance of the results.

### 2.1. Vane Profile and Cascade Parameters

The test article under consideration is the film-cooled LS-94 transonic turbine vane, formed by extruding its midspan profile: the LS-89, an uncooled base profile designed at the von Karman Institute for research purposes. The test model comprises a five-vane linear cascade, tested in the isentropic light-weight piston compression tube facility CT-2. The light-weight piston is powered by compressed air from a high-pressure tank to achieve the desired freestream Mach and Reynolds numbers. Additional information on this facility is available in [31]. Table 1 lists the geometrical dimensions of the cascade, while Figure 1 reports the related nomenclature and depicts the airfoil profile.

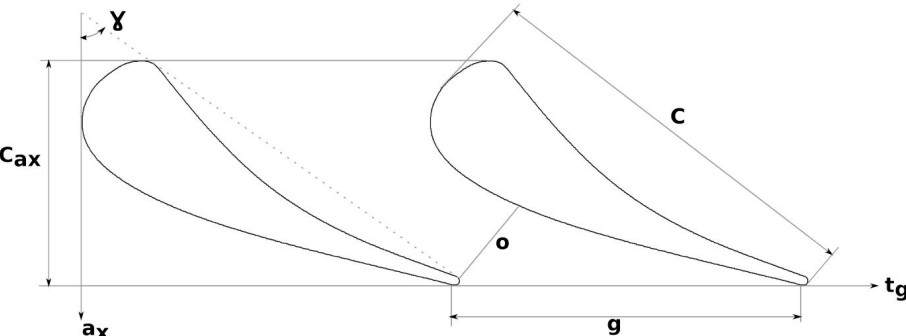

**Figure 1.** Cascade geometrical parameters: nomenclature (reproduced from Fontaneto [29]).

**Table 1.** Cascade geometrical parameters: description, symbols, and values.

| Parameter | Symbol | Value |
|---|---|---|
| Chord [mm] | $C$ | 67.65 |
| Axial chord [mm] | $C_{ax}$ | 36.985 |
| Stagger angle [°] | $\gamma$ | 55 |
| Throat [mm] | o | 14.93 |
| Pitch [mm] | g | 57.5 |
| Vane height [mm] | h | 100 |
| Trailing-edge radius [mm] | $r_{TE}$ | 4.13 |

### 2.2. Cooling System Configuration

The suction and pressure sides of the vane each feature two staggered rows of holes, specifically designed for optimal thermal regulation. Two distinct feeding chambers supply these holes, ensuring uniform and efficient distribution of the cooling medium. Table 2 details the cooling system's geometric characteristics, including specific measurements and arrangements of the holes, along with their respective angles and diameters, all critical for cooling efficiency. Additionally, Figure 2 complements this information by providing a cross-sectional view of the LS-94 airfoil, visually representing the strategic placement of the cooling holes within the vane's overall structure.

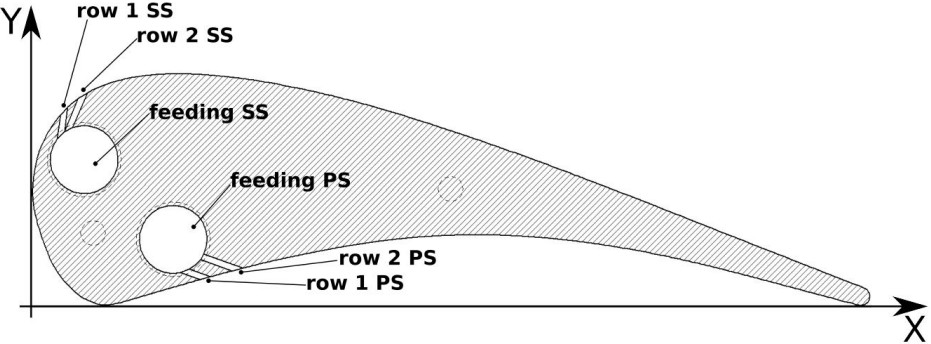

**Figure 2.** LS-94 cross-sectional view (reproduced from Fontaneto [29]).

**Table 2.** Cooling system characteristics.

| Parameter | Suction Side | | Pressure Side | |
|---|---|---|---|---|
| Diameter [mm] | 0.5 | | 0.5 | |
| Pitch [mm] | 1.5 | | 1.5 | |
| Coordinates [mm] | Row 1<br>X = 2.52<br>Y = 15.70 | Row 2<br>X = 4.05<br>Y = 19.95 | Row 1<br>X = 13.96<br>Y = 2.19 | Row 2<br>X = 16.59<br>Y = 2.90 |
| Angles [°] | Row 1<br>35° | Row 2<br>35° | Row 1<br>35° | Row 2<br>// to Row 1 |
| Number of holes | Row 1<br>33 | Row 2<br>32 | Row 1<br>33 | Row 2<br>32 |

*2.3. Experimental Configuration*

Experimental investigations were carried out to measure the vane velocity distribution, convective heat transfer, downstream pressure loss coefficient, and exit flow angle. Hot-wire anemometry technology facilitated the characterization of the inlet freestream turbulence. Specifically, a hot-wire (HW) probe was positioned at a distance of 148.7% of the axial chord length upstream of the leading-edge plane in the axial direction. Meticulous measurements of freestream turbulence intensity and spectrum were undertaken, with the freestream turbulence generated by a grid of staggered horizontal cylinder arrays strategically positioned upstream of the guide vane. To capture downstream total pressure wakes, a directional pneumatic pressure probe (PP) was traversed along the outlet measuring plane, precisely located at 43.3% of the axial chord length downstream from the trailing edge. All measurements were taken at midspan.

Additionally, comprehensive measurements of coolant total pressure $p_{0,c}$, total temperature $T_{0,c}$, and mass flow $\dot{m}_c$ were executed. The term "uncooled" is employed herein to designate tests conducted without coolant injection. Thanks to a valve, the cooling circuit was deliberately obstructed immediately upstream of the feeding chambers, causing the film-cooling holes to be unfed. Unlike some previous studies, such as the one conducted by Camci and Arts [32], where feeding chambers were filled to avoid the generation of spurious jets, in the present campaign, these feeding chambers were intentionally left empty. The heat transfer measurements were carried out along a Macor glass-ceramic vane surface coated with 42 platinum thin-film (TF) layers together with a semi-infinite body assumption to compute heat flux. The thin films designated for measurements were strategically positioned in the "clean" region, wherein the time-averaged flow manifested as two-dimensional. The experimental tests used a transient measurement technique exploiting the semi-infinite substrate assumption (Schultz et al. [33]), which expresses the convective heat transfer rate as a function of the substrate characteristics and the surface temperature evolution over time, which is actually what the thin film measures. Further details can be found in Fontaneto's works [29,30]. The static outlet pressure $p_2$ was determined using the static pressure taps rack. A concise summary of the measured quantities and the corresponding positions of the measuring points pertinent to the cascade is presented in Table 3 and Figure 3. The uncertainties associated with the various measurements were rigorously evaluated. Quantification revealed that the uncertainties for pressure, integrated loss coefficient, exit flow angle, and heat transfer coefficient were −0.5%, −0.2 points, −0.5°, and −5%, respectively.

**Table 3.** Measured quantities and corresponding positions.

| Section | $\%C_{ax}$ | Quantities |
|---|---|---|
| Inlet | 148.7%$C_{ax}$ upstream of leading edge | Turbulence (HW) $p_{01}$ $p_1$ $T_{01}$ |
| Outlet | 43.3%$C_{ax}$ downstream of trailing edge | Total pressure wakes (PP) $p_2$ |
| Coolant | / | $p_{0c}$ $T_{0c}$ $\dot{m}_c$ |

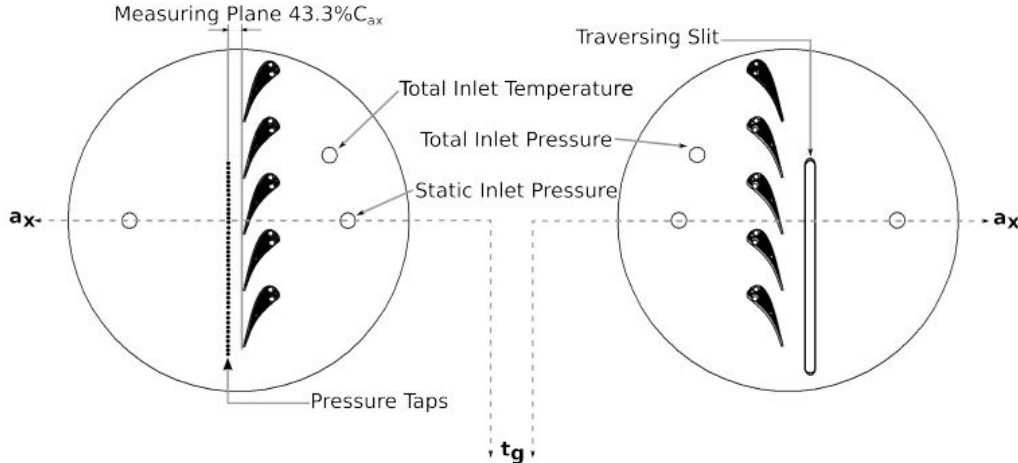

**Figure 3.** List of the features installed on the endwalls (reproduced from Fontaneto [29]).

## 3. Computational Methodology

This section presents the methodology used in the analysis. First, the test case is replicated in a CFD model, and the numerical setup is described in detail. Additionally, the results of the mesh sensitivity analysis are presented, which aided in choosing the appropriate computational domain for the turbulence transition model investigation activity.

### 3.1. Computational Domain

As described, the VKI test case consisted of a five-vane linear cascade. However, to effectively perform a CFD analysis with reasonably reduced computational cost and simulation time, only the mid-channel vane was considered. This domain simplification is helped by the fact that the aerothermal experimental measurements were performed only at the midspan of the middle vane, to be insensitive to endwall effects. Furthermore, since the tested geometry exhibits good spanwise periodic behavior, a single periodicity of 3 mm was considered to include one film-cooling hole per row both on the suction and pressure sides. The reduced domain is presented in Figure 4a, while Figure 4b displays the boundary conditions. Concerning the latter, translational periodicities were set in both the spanwise (Z) and radial (Y) directions.

### 3.2. Numerical Setup and Boundary Conditions

The commercial software Ansys Fluent 2023 R2 was used for the analysis. The compressible Navier–Stokes equations were solved, assuming the ideal gas law for the equation of state and a six-grade polynomial temperature-dependent formulation from NASA for the specific heat at constant pressure. The dynamic viscosity was treated with Sutherland's law, and the thermal conductivity was calculated using the kinetic theory. Momentum and

pressure-based continuity equations were solved together using the coupled algorithm to achieve a more robust convergence, and a second-order discretization was imposed for each quantity. The $k - \omega$ SST turbulence model [34] provided closure for the turbulent unknown terms. The model is suitable for both external aerodynamics and general-purpose CFD, performing well with separation and adverse pressure gradients [35]. Total pressure and total temperature injection conditions were applied at the inlet (based on experimental data), whereas a static pressure value at the outlet set the isentropic Mach number downstream. Following the experimental setup, the same computational domain was used for all cases, both with and without coolant injection. For the cooled cases, the desired coolant feeding chamber was triggered with a mass flow inlet condition, while in the other one, a wall condition was applied at the coolant inlet surfaces. Consequently, the untriggered film-cooling holes also affected the flow condition due to their presence, which is explained in Section 4.

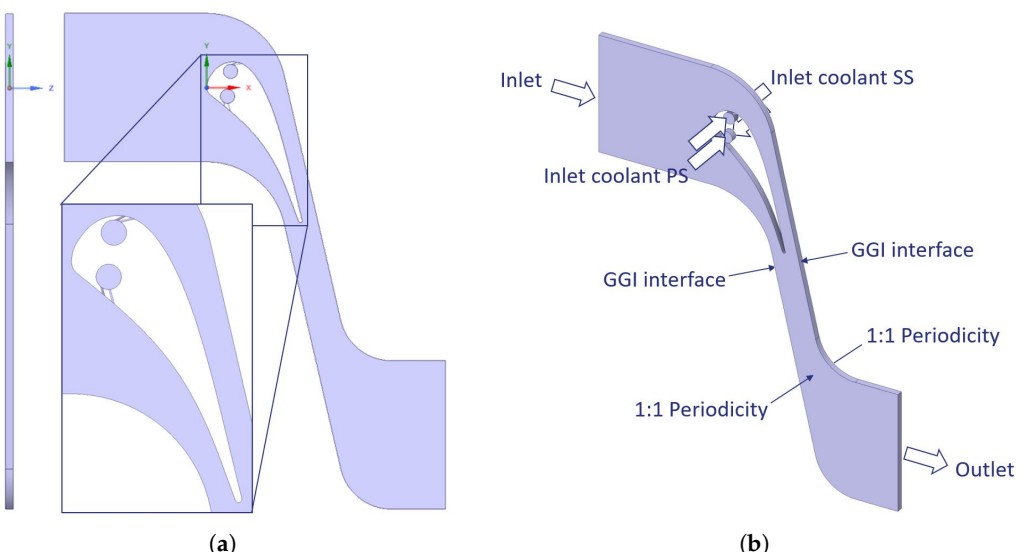

**Figure 4.** Three-dimensional computational domain of the VKI LS-94 test case: (**a**) 3 mm thick reduced domain and (**b**) boundary conditions.

A transported passive scalar $\Phi$ was used to track film coverage along the profile. A passive scalar refers to a scalar quantity that does not actively participate in the flow physics during a CFD simulation. This is only possible if the source of the passive scalar can be treated as a fluid with contaminants or species present in low concentration. These contaminants or species are transported along with the fluid flow and do not significantly affect the thermophysical properties of the fluid. Hence, in terms of boundary conditions, a unitary value was imposed at the coolant inlet, while a null value was prescribed at the mainstream inlet.

Concerning the boundary conditions for thermal characterization, in contrast to the experimental setup described in Section 2.3, a steady-state approach was chosen. This implies that the flow and thermal conditions remain constant over time in the CFD simulation. Hence, in the present analysis, the same initial temperature was applied to the surface of the vane, consistent with the experimental setup, but considering the experimental boundary conditions as a regime state. In the postprocessing phase, the heat transfer coefficient along the abscissa of the airfoil, utilizing a lateral average operation, is defined as:

$$HTC = \frac{q}{T_{wall} - T_{01}} \tag{1}$$

where $q$ is the wall heat flux, $T_{wall}$ is the wall temperature, and $T_{01}$ is the inlet total temperature.

Careful consideration was given to the inlet turbulence level $Tu_0$ and the eddy length scale value. Consistent with [29], where a turbulence generator grid was moved upstream

to the leading edge to vary the turbulence intensity level, this paper focused on the case where the generators were positioned closest at 202 mm upstream to the leading edge, corresponding to an inlet turbulence intensity of $Tu_{intensity} = 5.3\%$ and an eddy length scale of $Tu_{scale}$ = 7.5 mm. The latter represents the experimental integral length scale, which was used in the simulations as a turbulent length scale to determine k and omega. Concerning the lateral surfaces, a translational periodicity condition was enforced to simulate a linear vane cascade. The experimental investigation encompassed a broad spectrum of freestream conditions. Specifically, this paper delved into the analysis of three distinct configurations, denoted as test points $TF170113.001$ (uncooled), $TF280113.008$ (PS injection), and $TF170113.008$ (SS injection), as outlined in Table 4. The Reynolds number $Re$ under consideration was approximately $1.5 \times 10^6$, computed based on the chord length $C$ and the outlet velocity, corresponding to an outlet isentropic Mach number $Ma_{is,2}$ of 0.8.

**Table 4.** Details of the operating conditions for the examined test points.

| Test Case | $T_{01}$ [K] | $T_{0c}$ [K] | $p_{01}$ [Pa] | $p_2$ [Pa] | $p_{0c}$ [Pa] | $MF_c$ [g/s] | $Ma_{is,2}$ [-] | $Re$ [-] | $T_{wall,init}$ [K] | $Tu_{intensity}$ [-] | $Tu_{scale}$ [mm] |
|---|---|---|---|---|---|---|---|---|---|---|---|
| Uncooled | 414.3 | - | 262,674 | 168,501 | - | - | 0.853 | 1,627,122 | 290.7 | 5.3% | 7.5 |
| SS injection | 419.8 | 295.0 | 260,952 | 168,844 | 216,567 | 2.553 | 0.815 | 1,556,992 | 291.6 | 5.3% | 7.5 |
| PS injection | 423.5 | 271.6 | 262,995 | 168,891 | 285,062 | 3.651 | 0.822 | 1,558,704 | 292.5 | 5.3% | 7.5 |

### 3.3. Transition Modeling

The three models implemented in Ansys Fluent and used in the context of this work are described below.

#### 3.3.1. $k - k_l - \omega$ Transition Model

The $k - k_l - \omega$ transition model by Walters and Cokljat [3] is regarded as a three-equation eddy-viscosity type. It includes transport equations for turbulent kinetic energy $k$ (Equation (2)), laminar kinetic energy $k_l$ (Equation (3)), and the specific dissipation rate $\omega$ (Equation (4)).

$$\frac{Dk}{Dt} = P_k + R + R_{NAT} - \omega k - D_T + \frac{\partial}{\partial x_j}\left[\left(\nu + \frac{\alpha_T}{\alpha_k}\right)\frac{\partial k}{\partial x_j}\right] \tag{2}$$

$$\frac{Dk_l}{Dt} = P_{k_l} - R - R_{NAT} - D_l + \frac{\partial}{\partial x_j}\left[\nu\frac{\partial k_l}{\partial x_j}\right] \tag{3}$$

$$\frac{D\omega}{Dt} = C_{\omega 1}\frac{\omega}{k}P_k + \left(\frac{C_{\omega R}}{f_W} - 1\right)\frac{\omega}{k}(R + R_{NAT})$$
$$- C_{\omega 2}\omega^2 + C_{\omega 3}f_\omega\alpha_T f_W^2\frac{\sqrt{k}}{d^3} + \frac{\partial}{\partial x_j}\left[\left(\nu + \frac{\alpha_T}{\alpha_\omega}\right)\frac{\partial\omega}{\partial x_j}\right] \tag{4}$$

Given the complex formulation of the model, the authors refer the interested reader to [3] and the Ansys Fluent Theory Guide [36] for further details. The model is calibrated to reproduce the averaged effect of the breakdown of stream-wise fluctuations into turbulence during the bypass transition.

#### 3.3.2. $\gamma - Re_\theta$ Transition Model (Transition SST Model)

The $\gamma - Re_\theta$ transition model by Langtry and Menter [4], also named the Transition SST model in Ansys Fluent, is based on the coupling of the $k - \omega$ SST transport equations with two other transport equations: one for the intermittency $\gamma$ (Equation (5)) and another for the transition onset criteria in terms of the momentum-thickness Reynolds number

$Re_\theta$ (Equation (6)). This makes it a four-equation eddy-viscosity-type model. An ANSYS empirical correlation (Langtry and Menter [37]) has been developed to cover standard by-pass transition as well as flows in low-freestream-turbulence environments. The transition model can also be used in the presence of rough walls, thanks to a roughness correlation that requires the geometric roughness height as an input parameter and is applied as a corrective factor for the existing correlation for the transition momentum-thickness Reynolds number.

$$\frac{\partial(\rho\gamma)}{\partial t} + \frac{\partial(\rho U_j \gamma)}{\partial x_j} = P_{\gamma 1} - E_{\gamma 1} + P_{\gamma 2} - E_{\gamma 2} + \frac{\partial}{\partial x_j}\left[\left(\mu + \frac{\mu_t}{\sigma_\gamma}\right)\frac{\partial\gamma}{\partial x_j}\right] \tag{5}$$

$$\frac{\partial(\rho Re_\theta)}{\partial t} + \frac{\partial(\rho U_j Re_\theta)}{\partial x_j} = P_{\theta t} + \frac{\partial}{\partial x_j}\left[\sigma_{\theta t}(\mu + \mu_t)\frac{\partial Re_\theta}{\partial x_j}\right] \tag{6}$$

For further details, the authors refer the interested reader to the Ansys Fluent Theory Guide [36].

### 3.3.3. $\gamma$ Transition Model (Intermittency Model)

The $\gamma$ transition model by Menter et al. [6], also named the Intermittency transition model in Ansys Fluent, is a further development based on the $\gamma - Re_\theta$. This variant solves only one transport equation for turbulence intermittency $\gamma$ (Equation (7)) and avoids the need for the second $Re_\theta$ equation. This provides some advantages compared to the $\gamma - Re_\theta$ transition model, namely a reduction in computational effort, independence from the velocity field for $Re_\theta$ (thus making the model applicable to surfaces that move relative to the coordinate system for which the velocity field is computed), the ability to capture crossflow instability, and the possibility to fine-tune the model based on a small number of user parameters.

$$\frac{\partial(\rho\gamma)}{\partial t} + \frac{\partial(\rho U_j \gamma)}{\partial x_j} = P_\gamma - E_\gamma + \frac{\partial}{\partial x_j}\left[\left(\mu + \frac{\mu_t}{\sigma_\gamma}\right)\frac{\partial\gamma}{\partial x_j}\right] \tag{7}$$

For further details, the authors refer the interested reader to the Ansys Fluent Theory Guide [36].

### 3.4. Mesh Sensitivity

The fluid domain was discretized using an unstructured tetrahedral mesh. This mesh, depicted in Figure 5, features a global volume sizing of 1 mm and a surface sizing of 0.31 mm. Furthermore, 30 prismatic layers were applied to the walls to enhance simulation accuracy for the transition models. Moreover, three bodies of influence with refinements of 0.33 mm, 0.5 mm, and 0.75 mm were placed at the trailing edge to better discretize the wake. During the meshing process, particular attention was paid to ensuring a $y^+ < 1$ distribution over the entire surface. This requirement is crucial when dealing with aerothermal simulations over gas turbine vanes, particularly when using transition models. Moreover, following ANSYS best practices, the mesh expansion ratio in the wall-normal direction was kept under 1.2, and the aspect ratio of the near-wall cells was kept below 200 to ensure that the mesh was appropriate for the transitional models. A sensitivity analysis was conducted to ensure that the chosen mesh was the best candidate to efficiently capture the physics of the problem with optimal computational effort. In particular, the influence of the surface sizing and the number of layers was investigated for the uncooled, SS-cooled, and PS-cooled cases while keeping the height of the prismatic layer constant to discretize the boundary layer on the whole airfoil. Table 5 presents the characteristics of the mesh.

The choice of the final mesh was determined based on the results of the spanwise-averaged heat transfer coefficient along the profile. Positive values of curvilinear abscissa refer to the suction side, while negative ones refer to the pressure side. From Figures 6–8,

it is evident that the mesh adopted for the steady-state calculations is a good candidate for this work. The further refinement tested for injection cases with a surface mesh of 0.20 mm did not seem to provide any improvement. In particular, it should be noted that the most impactful factor on the analyzed meshes is the number of prism layers, rather than the surface sizing. Although not substantial, the differences with 30 layers are marked, especially regarding the smoothness of the profiles. In conclusion, the initial mesh with a surface sizing of 0.31 mm and 30 layers appeared to be a good candidate for the activity.

**Table 5.** Grid characteristics for the mesh sensitivity.

| Parameter | Non-Cooled | SS Injection | PS Injection |
|---|---|---|---|
| Global sizing [mm] | 1 | 1 | 1 |
| Surface sizing [mm] | 0.31<br>0.70<br>2.25 | 0.20<br>0.31 | 0.20<br>0.31 |
| Prismatic layers | 30 | 10<br>20<br>30 | 10<br>20<br>30 |

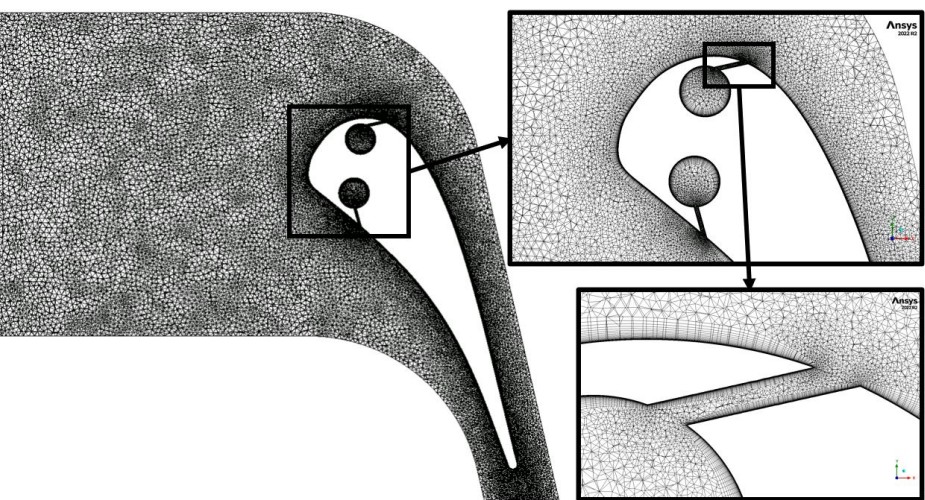

**Figure 5.** Computational mesh of the reduced domain for the steady flow simulation of the LS-94 vane, featuring a volume global sizing of 1 mm, surface sizing of 0.31 mm, and 30 prismatic layers. The figure depicts detailed views of the leading-edge region and one film-cooling hole on the suction side.

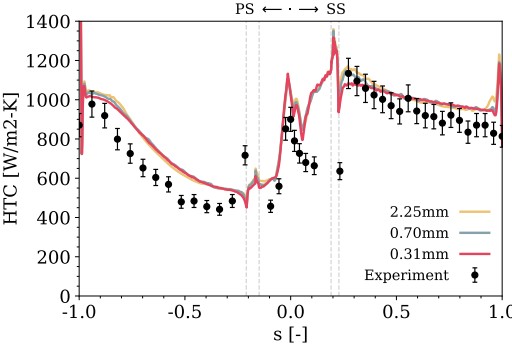

**Figure 6.** Mesh sensitivity results for the uncooled case using the fully turbulent $k - \omega$ SST RANS turbulence model, focusing on the surface sizing effect. Light-gray dashed lines indicate the positions of the film-cooling rows. Black dots represent the experimental results from Fontaneto [29], with corresponding 6.8% uncertainty level bars.

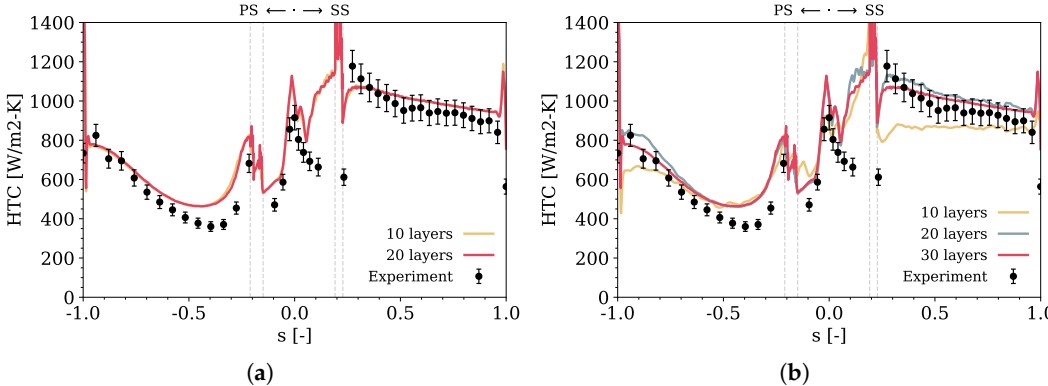

**Figure 7.** Mesh sensitivity results for the PS injection case using the fully turbulent $k - \omega$ SST RANS turbulence model. Light-gray dashed lines indicate the positions of the film-cooling rows. Black dots represent the experimental results from Fontaneto [29], with corresponding 6.8% uncertainty level bars. (**a**) Surface sizing effect. (**b**) Prismatic layers effect.

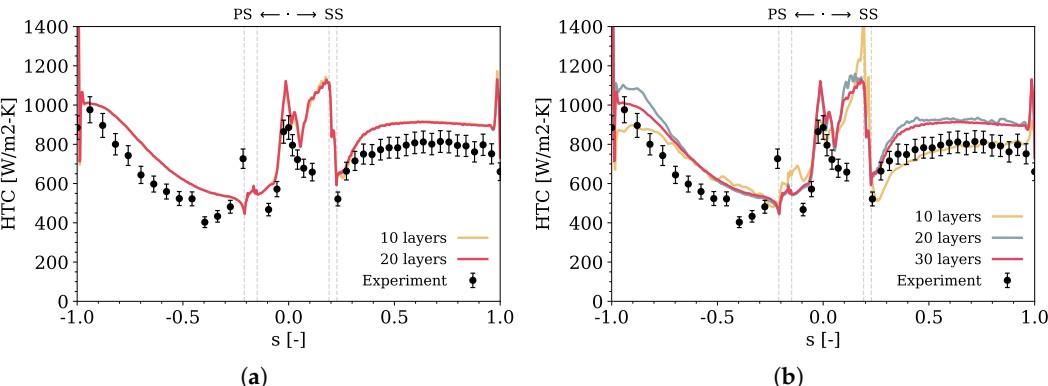

**Figure 8.** Mesh sensitivity results for the SS injection case using the fully turbulent $k - \omega$ SST RANS turbulence model. Light-gray dashed lines indicate the positions of the film-cooling rows. Black dots represent the experimental results from Fontaneto [29], with corresponding 6.8% uncertainty level bars. (**a**) Surface sizing effect. (**b**) Prismatic layers effect.

*3.5. Scale-Resolving Modeling*

To assess the capability of LES (or LES-like) approaches in predicting the aerothermal conditions on a film-cooled highly loaded gas turbine vane, it was decided to test a hybrid RANS/LES model. In the LS-94 setup, the presence of film cooling is a predominant source of unsteadiness, even in the absence of coolant injection, as a consequence of the disturbance generated by the holes. Accurately simulating vortex shedding is crucial for capturing the characteristics of unsteady flows and their effects on the boundary layers of the vane. LES would be the way to go when dealing with turbulent flows due to its independence from any turbulence model, as the subgrid-scale (SGS) model plays only a minimal role in determining turbulent mixing. However, a wall-resolved LES is impractically expensive due to the cost associated with the resolution of the near-wall region. A wall-modeled LES treatment, which combines the benefits of LES with a wall model to accurately predict the near-wall flow while reducing the computational cost, would undoubtedly be more effective for this test case. However, the laminar flow developing in the vicinity of the body's leading edge is not modeled when using this approach. Therefore, it should be completely resolved using the numerical method, imposing quite unaffordable grid requirements due to the very small thickness of the boundary layer in this region. Slotnick [38] estimated that the cost of accurately solving the very thin laminar boundary layer near the leading edge of an airfoil may be 10 to 100 times higher than the cost of solving the modeled turbulent region. Since the thickness of the developing laminar boundary layer in this region is very

small, the computational costs become prohibitive. Considering that RANS models are well suited for wall boundary layers, in recent decades, many hybrid RANS/LES models have been developed, starting from the Detached Eddy Simulation (DES) approach proposed by Spalart [39]. The Stress-Blended Eddy Simulation (SBES) model from Menter [40] is one of the latest developments that improve the formulation of the shielding function to switch between RANS and LES. This function crucially determines the application of RANS or LES in varying flow regions, ensuring accurate turbulence modeling and resolution.

$$v_t^{SBES} = f_s \cdot v_t^{RANS} + (1 - f_s) \cdot v_t^{LES}. \tag{8}$$

To avoid utilizing the LES SGS model throughout the entire flow field, a shielding function was implemented. This function ensured that a RANS approach was maintained only in selected zones, thus allowing for a lower mesh resolution and significantly reducing computational demands. As for the modeling approach, the $k - \omega$ SST turbulence model was used for the RANS component, while the WALE model from Nicoud and Ducros [41] was applied for the subgrid-scale in the SBES simulation.

To allow for a proper generation and transport of the turbulent eddies, some actions were undertaken. Firstly, the Synthetic Turbulence Generator by Shur et al. [42] was used at the inlet, and the height of the reduced domain was increased from 3 mm to 9 mm (considering a turbulent length scale of 7.5 mm). In addition, the mesh was refined both globally and locally. The global mesh size was reduced from 1.0 mm to 0.5 mm, while in the vicinity of the film-cooling holes, a body of influence with a refinement of 0.05 mm permitted a finer discretization for the hot gas–coolant interaction. Finally, it was possible to achieve a mesh consisting of 26.6 million elements.

### 3.6. SBES Quality

To assess the quality of the SBES simulation, it was decided to compute the output field named LES Resolution Quality, which represents the resolved portion of the turbulence kinetic energy, defined as:

$$q = k_{res}/(k_{res} + k_{mod}) \tag{9}$$

where $k_{res}$ is the resolved turbulence kinetic energy and $k_{mod}$ is the modeled part. The resolved kinetic energy is directly computed from the mean square velocity fluctuations:

$$k_{res} = 0.5(\overline{u'u'} + \overline{v'v'} + \overline{w'w'}) \tag{10}$$

and the time-averaged modeled subgrid-scale kinetic energy is evaluated based on the subgrid-scale eddy viscosity $\mu_t$, density $\rho$, and strain-rate magnitude $S$:

$$k_{mod} = \overline{\frac{1}{3} \frac{\mu_t}{\rho} S} \tag{11}$$

According to Pope [43], who proposed this criterion, an LES Resolution Quality of 0.8 or higher typically indicates a sufficiently resolved turbulent flow solution. The left side of Figure 9 depicts the LES Resolution Quality and highlights that the mesh quality is satisfactory across the whole computational domain. The distribution drops below 0.8 only in the vicinity of the walls, but this is not a concern since the right side of Figure 9 shows that the very same region is characterized by a shielding function value equal to one (red), meaning that the solution is resolved by RANS.

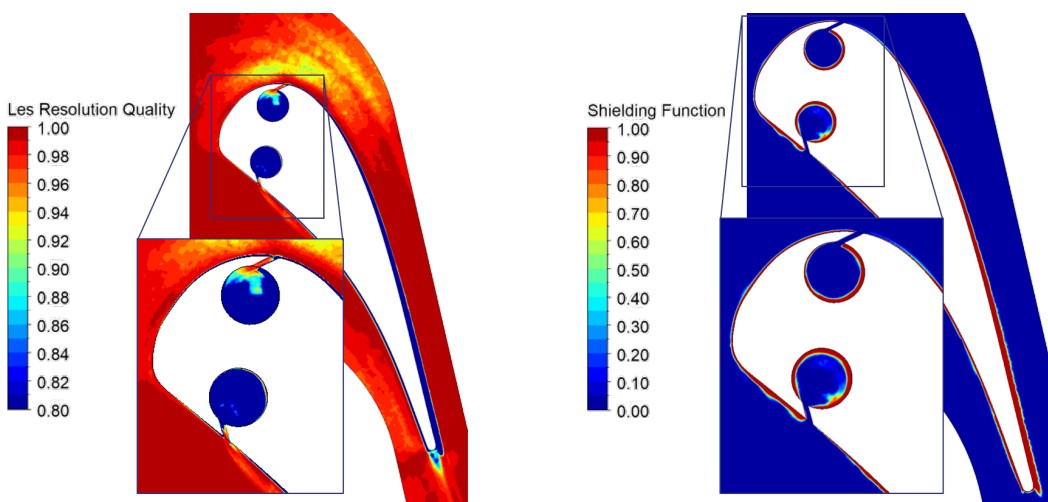

**Figure 9.** Contours of the LES Resolution Quality and shielding function fields for the SBES quality assessment of the PS injection test case.

## 4. Results

This chapter presents the key findings of this study, offering detailed comparisons between various simulation models in predicting the heat transfer coefficient. The results are structured to highlight significant trends and discrepancies, providing a basis for further analysis and discussion of the research implications.

### 4.1. Assessment of Transition Models

This study compared the results from a fully turbulent simulation using the $k - \omega$ SST RANS turbulence model with those from transition-sensitive simulations employing the previously outlined $\gamma$ and $\gamma - Re_\theta$ transition models. This analysis encompassed uncooled, PS injection, and SS injection cases to deepen the understanding of model performance on turbine vanes under film-cooling conditions.

#### 4.1.1. Uncooled

Figure 10 depicts the uncooled working condition of the vane, computed using the fully turbulent $k - \omega$ SST RANS model. The Mach number, static pressure, and density gradient (Schlieren) fields are shown on the left, center, and right of the figure.

Before proceeding to the thermal analysis, it is essential to ensure that the different numerical approaches adopted in this study yield the same mean aerodynamic flow, which is a prerequisite for comparing the aerothermal behavior. Figure 11 depicts the isentropic Mach number plotted against the non-dimensional curvilinear abscissa for the uncooled vane test case. As no differences can be observed among the various models in the figure, the analysis can proceed.

Figure 12 shows the HTC distribution on the airfoil for the uncooled configuration. It can be observed that at the leading edge, all three transition models accurately predict the experimental HTC values, within or very close to the nominal 6.8% uncertainty. The fully turbulent result slightly overpredicts the values but they are acceptable at the stagnation point. However, the solution then rapidly diverges from the experimental data and the transitional prediction when it approaches the film-cooling holes. Especially on the highly curved region of the SS before the two rows ($0.10 < s < 0.20$), the fully turbulent model returns a significantly higher prediction of the HTC. This can be attributed to the strong acceleration of the flow, which reduces the thickness of the turbulent BL and increases the temperature gradient in the vicinity of the wall. Moving further downstream, there is a noticeable difference in the performance of the models. On the pressure side, the fully turbulent model better follows the experimental data with a slight overestimation of the HTC, while all the transition models significantly underestimate it, especially toward the trailing edge. This seems to indicate

that the transition models maintain a laminar condition for the BL after the film-cooling holes, located in the range $-0.25 < s < -0.20$, while the experimental evidence suggests the presence of the two rows may trigger a transition. This behavior is consistent with other results from the literature [21,24], where a similar heat transfer underestimation was observed on the PS with the transitions models for inlet turbulence intensity between 4 and 6%. On the suction side, the presence of the film-cooling holes ($s = 0.20$–$0.25$) induces the transition from a laminar to a turbulent boundary layer and the subsequent increase in the heat transfer coefficient. This behavior is well characterized by the $\gamma$ and $\gamma - Re_\theta$ transition models, while the $k - k_l - \omega$ model seems to maintain the BL in a laminar status, roughly until $s = 0.40$. The fully turbulent solution, which cannot reproduce this phenomenon, results in a general overestimation up to $s = 0.25$. After the transition, all models tend to slightly overestimate the HTC, with the fully turbulent model being a better match with the experimental data, remaining within the 6.8% uncertainty range [29]. The transition models seem to overpredict the turbulence generation, which can be attributed to the BL transition, ultimately resulting in excessive heat transfer.

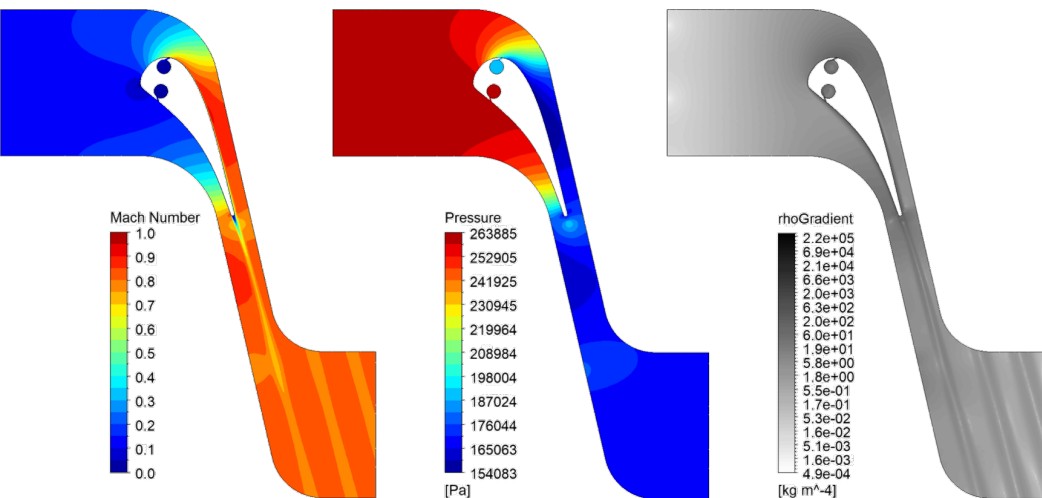

**Figure 10.** Fully turbulent $k - \omega$ SST RANS Mach number (**left**), static pressure (**center**), and Schlieren (**right**) contours for the uncooled vane.

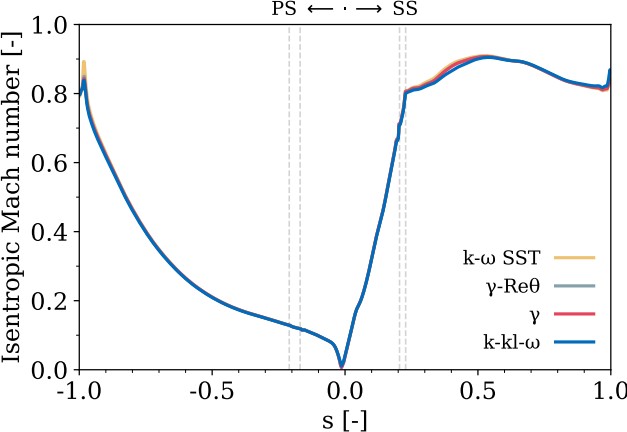

**Figure 11.** RANS spanwise-averaged isentropic Mach number results for the uncooled vane plotted against the non-dimensional curvilinear abscissa. Light-gray dashed lines indicate the positions of the film-cooling rows.

To understand the different *HTC* distributions between the two approaches for transition modeling, it was decided to plot the decay of the turbulence intensity (see Figure 13). As can be observed, the fully turbulent model and the $\gamma - Re_\theta$ transition model return roughly the same distribution, with small differences on the second part of the suction

side. However, the $k - k_l - \omega$ model shows a significant discrepancy. Despite having the same boundary condition at the inlet in terms of the turbulence intensity $Tu_{intensity}$ and the integral length scale $Tu_{scale}$, it shows a steep decay compared to the $\gamma - Re_\theta$ model. Consequently, the turbulence intensity at the leading edge of the vane is significantly lower, leading to a lower HTC and a lower propensity to trigger the transition of the laminar BL to a turbulent condition.

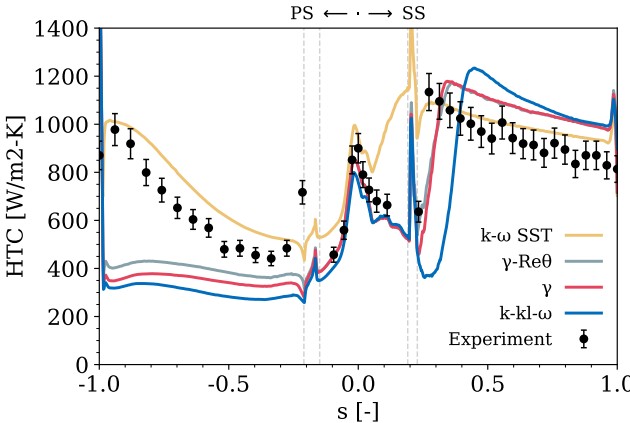

**Figure 12.** Experimental and RANS spanwise-averaged heat transfer coefficient results for the uncooled vane plotted against the non-dimensional curvilinear abscissa. Light-gray dashed lines indicate the positions of the film-cooling rows. Experimental data were obtained from thin-film thermometry measurements along the vane midspan by Fontaneto [29], with corresponding 6.8% uncertainty level bars.

Figure 14 further clarifies some of the motivations behind the CFD results. The contours of the heat transfer coefficient on the vane surface and intermittency in a perpendicular plane offer a comprehensive view of the aerothermal behavior in the uncooled case using the $\gamma$ transition model (the same applies to the $\gamma - Re_\theta$). On the suction side, the transition model depicts a clear shift from a laminar to a turbulent boundary layer, as evidenced by the intermittency factor, which transitions from 0 (laminar flow) to 1 (fully turbulent flow). The laminar flow on the suction side initially exhibits lower HTC values due to its smoother and less chaotic nature. As the boundary layer transitions to turbulence, a concurrent rise in the HTC reflects the enhanced convective heat transfer associated with turbulent flows. In contrast, on the pressure side of the vane, the model indicates a persistent laminar boundary layer, as suggested by the intermittency factor remaining at or near 0 with no prominent switches. In this region, the flow does not undergo the laminar-to-turbulent transition observed on the suction side, so the HTC on the pressure side remains relatively low, consistent with the characteristics of a laminar flow, which typically presents less vigorous mixing and lower heat transfer rates compared to turbulent flow.

The results also reveal an intriguing phenomenon characterized by a distinct pattern on the CFD heat transfer coefficient profiles in the vicinity of the film-cooling holes on both the PS and SS. This pattern manifests as an initial sharp increase in the HTC, followed closely by a sudden decrease. This behavior is illustrated in Figure 15, providing insight into the underlying mechanism. This unusual pattern arises due to the specific configuration of the coolant chambers and the chosen modeling strategy. In this uncooled scenario, where the chambers are not supplied with coolant, the mainstream hot gas is ingested into the first row of film-cooling holes, leading to the increase in the HTC due to the suction of the BL and its subsequent renewal [44]. Once inside the cavity, the hot gas experiences a cooling effect within the internal plenum, which has the same wall temperature as the airfoil. Subsequently, the cooled gas is re-injected through the second row of film-cooling holes, generating a limited film-cooling effect that reduces the wall heat flux and, consequently, the HTC. It is worth pointing out that this effect might be attributable to the steady-state approach chosen to model the experimental test, which is instead transient in nature. It

is possible that, in real experimental conditions, the fast transient (roughly 0.15 s) and the lung effect provided by the coolant plenum would not behave in the way shown by steady-state CFD, which ultimately returns a regime solution.

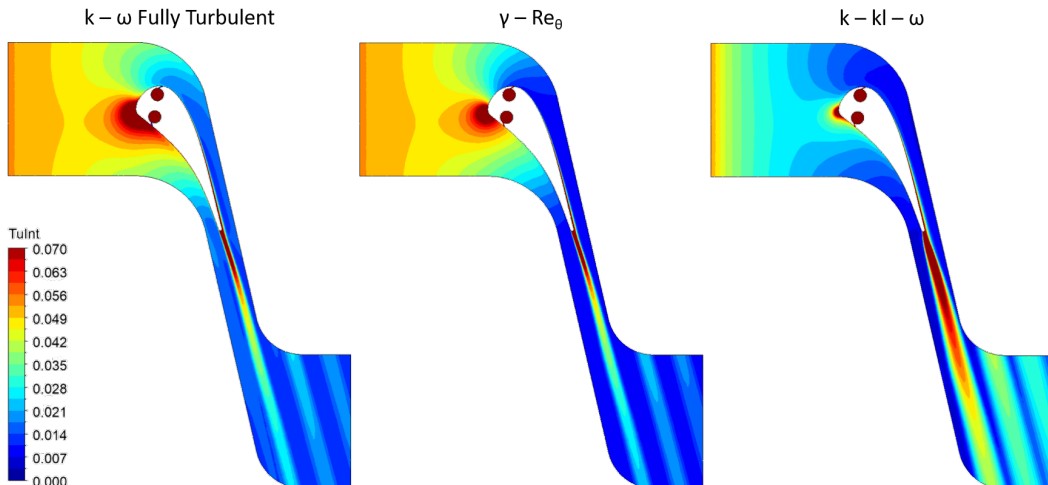

**Figure 13.** Decay of turbulence intensity from the inlet for the different turbulence transition models: fully turbulent $k - \omega$ SST model (**left**), $\gamma - Re_\theta$ model (**center**), and $k - k_l - \omega$ model (**right**). The $\gamma$ model is not depicted as it performed similarly to the $\gamma - Re_\theta$ model.

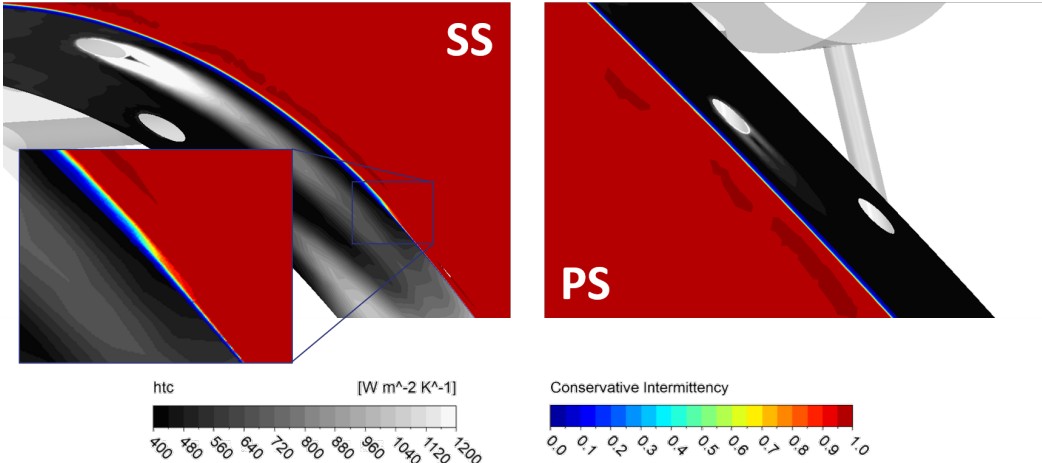

**Figure 14.** Heat transfer coefficient (vane surface) and intermittency (perpendicular plane) contours for the uncooled case using the $\gamma$ transition model. On the suction side (SS), the model switches from a laminar (intermittency = 0) to a turbulent (intermittency = 1) boundary layer, whereas on the pressure side (PS), it does not.

### 4.1.2. PS Injection

In the scenario with pressure-side injection, the results are quite similar to those already shown without injection. Figure 16 shows that the experimental HTC distribution remains unchanged at the leading edge and the suction side, whereas the heat transfer coefficient is reduced on the pressure side as a consequence of the coolant injection. On the PS, the minimum HTC at $s \approx 0.35$ is reduced from $\approx 450$ W/m$^2$K for the uncooled case to $\approx 350$ W/m$^2$K for the injection, whereas the maximum HTC is reduced from $\approx 1000$ W/m$^2$K to $\approx 850$ W/m$^2$K. Such a widespread effect on the PS can be attributed to the high blowing ratio of the film-cooling holes (3.455), which justifies penetration of the cooling into the freestream. The relatively high coolant mass flow rate (compared to the SS injection case) thus preserves its concentration, providing some cooling effect even approaching the trailing edge.

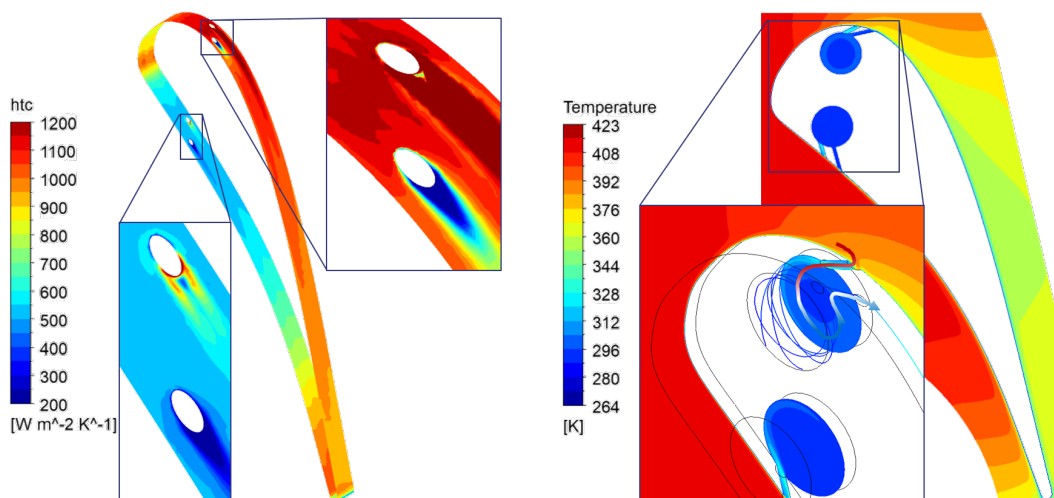

**Figure 15.** Heat transfer coefficient and temperature contours for the uncooled case underlining the mainstream hot gas ingestion and re-injection behavior.

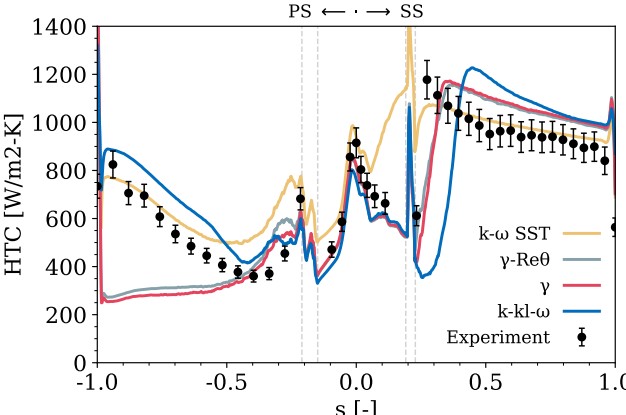

**Figure 16.** Experimental and RANS spanwise-averaged heat transfer coefficient results for the PS injection vane plotted against the non-dimensional curvilinear abscissa. Light-gray dashed lines indicate the positions of the film-cooling rows. Experimental data were obtained from thin-film thermometry measurements along the vane midspan by Fontaneto [29], with corresponding 6.8% uncertainty level bars.

As far as the CFD results are concerned, the same considerations made for Figure 12 apply to the leading-edge region. There, the presence of the film-cooling rows determines a strong difference between the two approaches for transition modeling. $\gamma - Re_\theta$ is generally better at predicting transition on the SS, despite some overestimation in the HTC for $s > 0.40$. On the PS, the situation did not change much compared to the uncooled case, with a BL that seems to maintain its laminarity until the trailing edge. In contrast, the $k - k_l - \omega$ model is still deficient in capturing the transition on the SS (due to the above-mentioned excessive decay in the turbulence intensity). However, quite surprisingly, for the PS-injection case, a transition of the BL for $s < -0.45$ can be observed. This ultimately leads to a divergence between the two modeling approaches and an overestimation of the predicted HTC concerning the experimental value. Finally, the $k - \omega$ SST model consistently overestimates in the region of the leading edge, but overall, it offers a decent prediction downstream of the film-cooling rows, i.e., along most of the profile, which represents a conservative (and thus appealing) approach in the design phase.

The detailed analysis of the results, as referenced in Figure 17, reveals that the coolant injected on the PS exhibits behavior consistent with a "penetration regime". In this regime, the coolant is ejected from the holes with enough momentum to penetrate the mainstream flow. Unlike scenarios where the coolant sticks closely to the surface (mass addition regime),

in this regime, the coolant does not immediately adhere to the vane surface after exiting the cooling holes. Instead, with higher momentum, it penetrates further into the mainstream flow before attaching to the surface. The magnified image of the heat transfer coefficient contour on the pressure side of the turbine reveals the presence of horseshoe vortex wakes, which are typical of this regime. This ultimately increases the HTC in the vicinity of the injection region. Since the vane behaves similarly to the uncooled case on the SS, the same effect of ingestion and re-injection of the hot mainstream gas occurs, as described in the previous section.

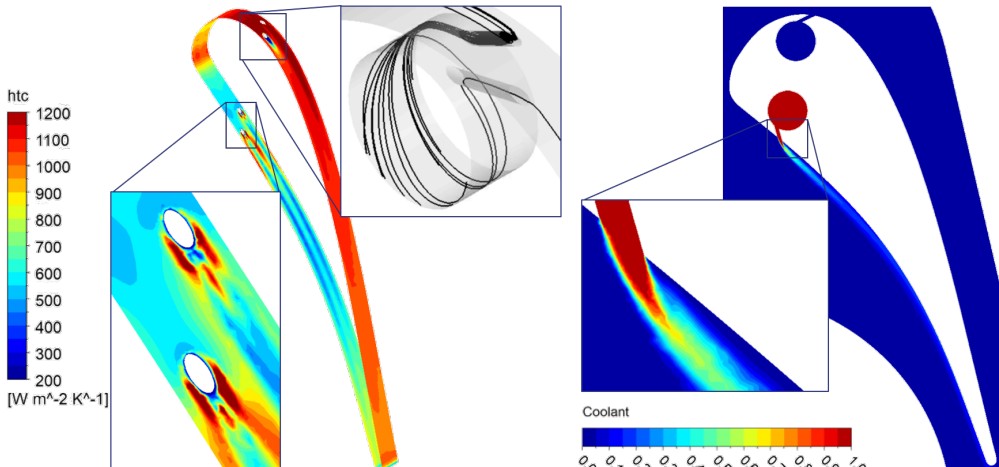

**Figure 17.** Heat transfer coefficient (**left**) and coolant concentration (**right**) contours for the PS injection case. The mainstream hot gas ingestion and re-injection phenomena are visible from the streamlines on the suction side and their effect on the HTC. The penetration of the coolant in the freestream and the poor coverage on the PS are visible from the coolant concentration contours.

### 4.1.3. SS Injection

Similarly to the observations made with the pressure side injection, injecting on the suction side of the vane also results in a noticeable reduction in the heat transfer coefficient profile within this area (see Figure 18). On the SS, the minimum HTC is reduced from $\approx$650 W/m$^2$K for the uncooled case to $\approx$500 W/m$^2$K for the injection, whereas the peak HTC of $\approx$1150 W/m$^2$K is completely smoothed away. However, approaching the TE ($s \approx 0.9$), a similar HTC value is shown for the two test points. This effect can be attributed to the low blowing ratio of this test point (0.454), which determines a mass addition regime for the coolant injection. Overall, this condition typically provides very good protection in the immediate vicinity of the holes, while the coverage effect steeply drops downstream due to the limited amount of coolant injected and its mixing with the freestream.

As far as the CFD results are concerned, as expected, no differences can be observed on the LE and the PS compared to the uncooled case. Interestingly, on the SS, all the CFD models show a reduction in the HTC, as observed in the experiments immediately after the coolant injection. The minimum value of the HTC is correctly realized at $s \approx 0.25$, except for the $k - k_l - \omega$ model, which significantly delays the transition, as in the uncooled case. This is somewhat unexpected because the PS injection case showed a triggering effect due to the coolant injection. Such a difference may be attributed to the different regimes of the film cooling, which, in case of mass addition, is not able to disturb the BL enough to trigger the transition in the $k - k_l - \omega$ model. Again, the $k - \omega$ SST model demonstrates a tendency to overestimate the heat transfer coefficient. However, compared to the other cases, the predictive performance shows a slightly increased discrepancy from the experimental data. This is indeed due to the presence of film cooling, which increases the complexity of the flow physics and the uncertainty associated with modeling the interaction between film-cooling jets and the mainstream.

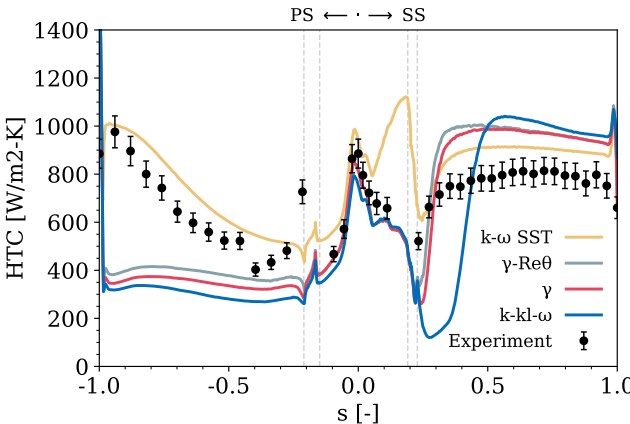

**Figure 18.** Experimental and RANS spanwise-averaged heat transfer coefficient results for the SS injection vane plotted against the non-dimensional curvilinear abscissa. Light-gray dashed lines indicate the positions of the film-cooling rows. Experimental data were obtained from thin-film thermometry measurements along the vane midspan by Fontaneto [29], with corresponding 6.8% uncertainty level bars.

In the context of coolant injection, in this case, the behavior of the coolant aligns with what is described as the "mixing regime". In this regime, the coolant rapidly mixes with the hot gas stream after being ejected from the cooling holes. Unlike the penetration regime, where the coolant maintains a distinct, separate flow path before gradually mixing with the mainstream flow, the mixing regime is characterized by a more immediate and thorough integration of the coolant with the surrounding hot gas. This results in a more homogenized temperature distribution around the cooling holes. Figure 19 provides a clear visual representation of this description, with the coolant adhering closely to the surface of the turbine blade after injection. Furthermore, the ingestion and re-injection phenomena due to the absence of film cooling on the pressure side are visible.

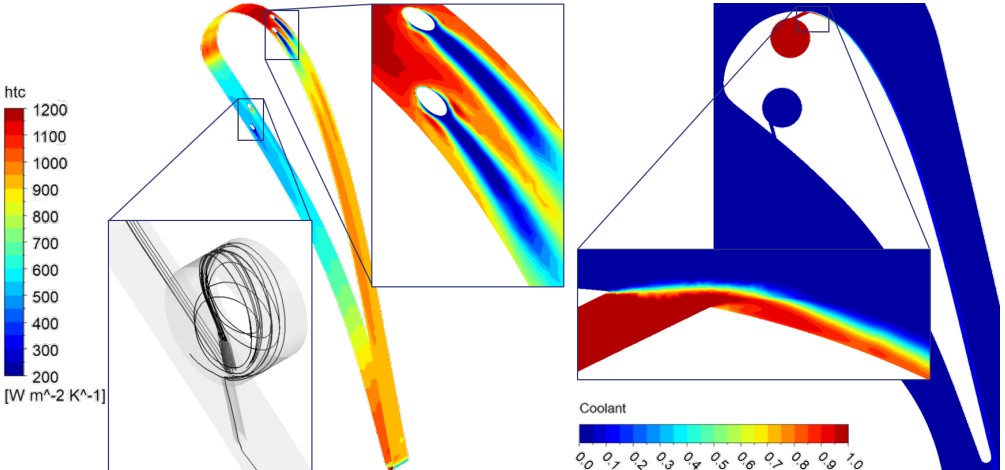

**Figure 19.** Heat transfer coefficient (**left**) and coolant concentration (**right**) contours for the SS injection case. The mainstream hot gas ingestion and re-injection phenomena are visible from the PS streamlines and their effect on the HTC. The limited penetration of the coolant and the good coverage close to the holes are visible from the coolant concentration contours.

### 4.2. Assessment of RANS vs. SBES

This section presents a comparative analysis between the RANS and SBES approaches on turbine vanes under film-cooling conditions, encompassing PS and SS injection scenarios to understand the differences between these models and the potential improvements associated with a transient approach. Similar to the previous analysis, it is fundamental to assess the impact on the aerodynamic performance of the vane. For this purpose, as a

preliminary check, the isentropic Mach number as a function of the curvilinear abscissa was extracted for RANS and SBES and is depicted in Figure 20. As expected, the impact of the transient modeling is negligible from this point of view.

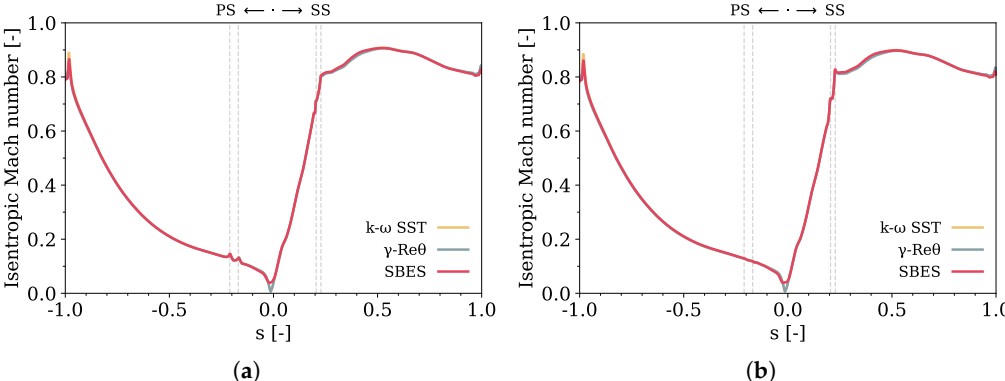

(a)

(b)

**Figure 20.** Spanwise-averaged isentropic Mach number results for the (**a**) PS injection and (**b**) SS injection cases plotted against the non-dimensional curvilinear abscissa. Light-gray dashed lines indicate the positions of the film-cooling rows.

### 4.2.1. PS Injection

To gain a better insight into the impact of transient modeling on the flow field, we can refer to Figure 21, which depicts the vorticity magnitude and the velocity curl in the Z-direction for the two cases. It can be observed that for RANS, almost no vorticity is generated at the inlet, with its generation starting to build up upon reaching the vane. The maximum values can be observed in the boundary layer and the wake created at the airfoil's trailing edge. In contrast, for the SBES case, vorticity is present even at the inlet due to the turbulence generated at the boundary. Similar to RANS, the maximum vorticity is achieved in the vicinity of the walls and downstream of the TE, with the difference being that the transient case allows for the depiction of the instantaneous field generated by velocity fluctuations. In addition, the velocity curl in the Z-direction was used to highlight the unsteadiness generated by the shear layer vortices due to the interaction between injected film cooling and the mainstream.

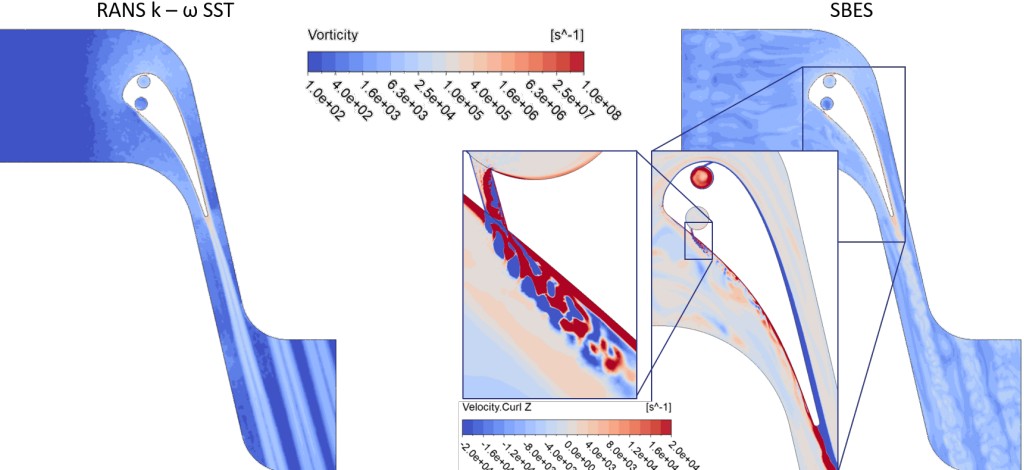

**Figure 21.** Vorticity magnitude for PS injection case for RANS (**left**) and SBES (**right**). A detailed view of the velocity curl in the Z-direction is provided for the SBES in the vicinity of the PS film-cooling holes.

Figure 22 depicts the span-averaged heat transfer coefficient and coolant mass fraction, i.e., the coverage results for the PS injection case. At the leading edge, the SBES returns a

result aligned with the results of the transition models rather than with the fully turbulent solution. In particular, the SBES prediction of the HTC is slightly below the experimental uncertainty. On the first part of the suction side, the SBES also fails to accurately predict the position of the laminar-to-turbulent transition. It follows the fully turbulent solution from $s \approx 0.1$ and overestimates the HTC before reaching the position of the film-cooling holes. This can be attributed to the absence of any model to address the laminarization of the boundary layer. Again, the HTC spikes symbolize the hot gas mainstream ingestion and re-injection phenomenon. Toward the trailing edge, the SBES solution falls between the RANS models, providing an estimation of the heat transfer coefficient that lies just within the uncertainty range.

In contrast, on the pressure side, the situation is different. Firstly, the deceleration zone that showed laminar behavior is well captured by the SBES until the position of the cooling holes. Immediately downstream of the FC holes ($s \approx -0.25$), the SBES solution does not overshoot but rather follows the slope of the experiment, unlike the $k - \omega$ SST and $\gamma - Re_\theta$ models. In Figure 22b, which displays the coolant coverage, this is demonstrated by the fact that for $-0.25 < s < -0.20$, the coverage is slightly higher with the SBES, indicating that the coolant was more evenly distributed in this region. After this region, a reversal of this behavior occurs: the SBES consistently overestimates the HTC by approximately $\approx$100–200 W/m$^2$K over the remaining region but better aligns with the slope of the experiment. Again, the coolant mass fraction chart explains this phenomenon by showing that for $s < -0.25$, the SBES exhibits lower coverage compared to the RANS approaches, which instead show similar trends. This behavior, characterized by a coverage peak far from the holes, can be explained by the high blowing ratio of the film cooling (3.455), which caused the coolant to penetrate the mainstream and provide some protection only downstream due to mixing.

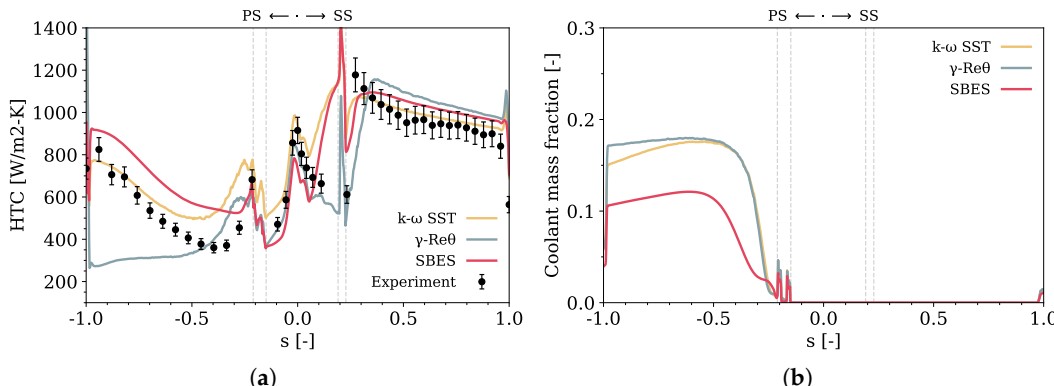

**Figure 22.** Experimental, fully turbulent $k - \omega$ SST RANS, $\gamma - Re_\theta$ transition model, and SBES heat transfer coefficient and coolant mass fraction results for the PS injection case plotted against the non-dimensional curvilinear abscissa. Light-gray dashed lines indicate the positions of the film-cooling rows. Experimental data were obtained from thin-film thermometry measurements along the vane midspan by Fontaneto [29], with corresponding 6.8% uncertainty level bars. (**a**) Heat transfer coefficient. (**b**) Coolant mass fraction.

Figure 23 further illustrates the described phenomenon by depicting the contours of the heat transfer coefficient (left) and coolant mass fraction (right) on the pressure side for this test case, comparing the SBES solution with the $k - \omega$ SST RANS. From the contours, it is clear that the differences in the HTC profiles downstream of the holes can be attributed to the prediction of the coverage, as a lower HTC is associated with higher coolant coverage. Despite the higher heat transfer coefficient value on the pressure side, the SBES better captures the slope of the experiment compared to the RANS solutions. Indeed, it would be interesting to have some experimental measurements of the film coverage to support the

validation of the CFD models and to discern how much the returned HTC was affected by the prediction of film cooling.

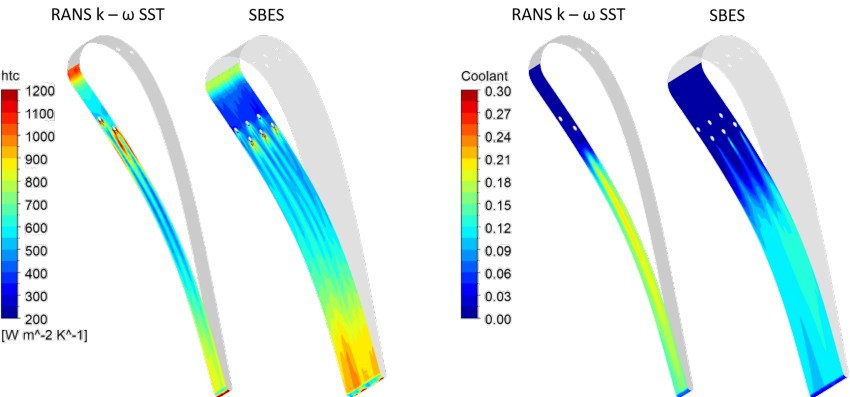

**Figure 23.** Heat transfer coefficient (**left**) and coolant mass fraction (**right**) contours for PS injection case for the different simulation approaches: steady-state fully turbulent $k - \omega$ SST RANS versus unsteady scale-resolving SBES modeling.

### 4.2.2. SS Injection

In the same way as for the PS injection case, Figure 24 depicts the results of the aerothermal simulations for the case with SS injection. In particular, Figure 24a depicts the span-averaged heat transfer coefficient distribution over the entire airfoil profile, whereas Figure 24b shows the span-averaged coolant mass fraction distribution. The results remain consistent at the leading-edge region, which is insensitive to the location of film-cooling injection. However, unlike the previous case, the SBES provides a more reliable representation of the thermal loads compared to the RANS models, both on the pressure and suction sides. The improvement in the solution on the suction side can be attributed to the film coverage profile. In fact, the more accurate prediction of the HTC (which is lower than RANS) results from a higher coolant concentration, especially in the vicinity of the film-cooling holes. Indeed, the transient LES-like modeling proves effective in spreading the coolant in the spanwise direction and overcoming the typical underestimation of turbulent mixing seen in RANS models.

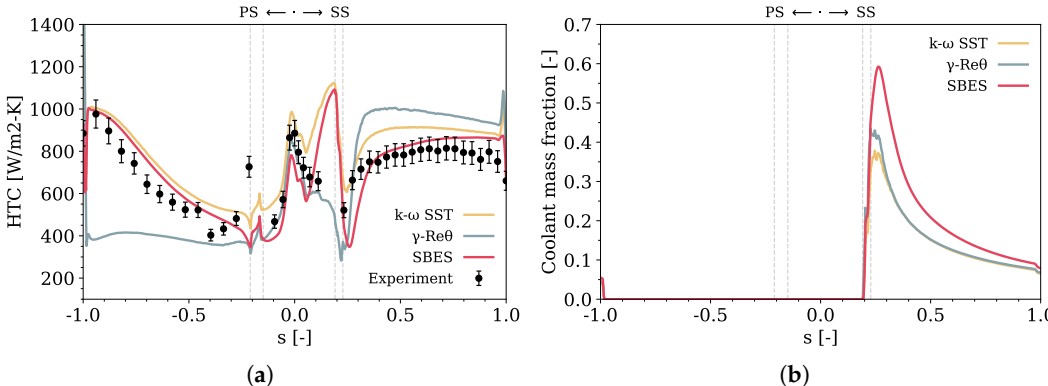

**Figure 24.** Experimental, fully turbulent $k - \omega$ SST RANS, $\gamma - Re_\theta$ transition model, and SBES heat transfer coefficient and coolant mass fraction results for the SS injection vane plotted against the non-dimensional curvilinear abscissa. Light-gray dashed lines indicate the positions of the film-cooling rows. Experimental data were obtained from thin-film thermometry measurements along the vane midspan by Fontaneto [29], with corresponding 6.8% uncertainty level bars. (**a**) Heat transfer coefficient. (**b**) Coolant mass fraction.

As a qualitative confirmation, Figures 24b and 25 show that SBES exhibits higher coverage compared to RANS modeling due to a more pronounced coolant spreading

throughout the vane. In contrast to the PS injection case, where the coverage gap between SBES and RANS remained nearly constant for $s < -0.35$ (Figure 22b), in this case, the gap is reduced moving toward the trailing edge due to the lower blowing ratio (0.454) and the consequent different regime of the film cooling, which provided excellent protection near the holes but a steeper decrease downstream due to insufficient mass flow rate and the mixing with the mainstream. Additionally, since the discrepancy in coverage between the $k - \omega$ SST and $\gamma - Re_\theta$ models is small, if not present, it can be concluded that the differences in the *HTC* profile between these two models can be attributed to the laminar/turbulent nature of the BL rather than the coolant coverage. On the other hand, since the discrepancy in coverage between the $k - \omega$ SST model and the SBES approach is significant, and since they both neglect the potential laminarization, it can be concluded that the differences in the *HTC* profile between these two models can be attributed to the coverage prediction.

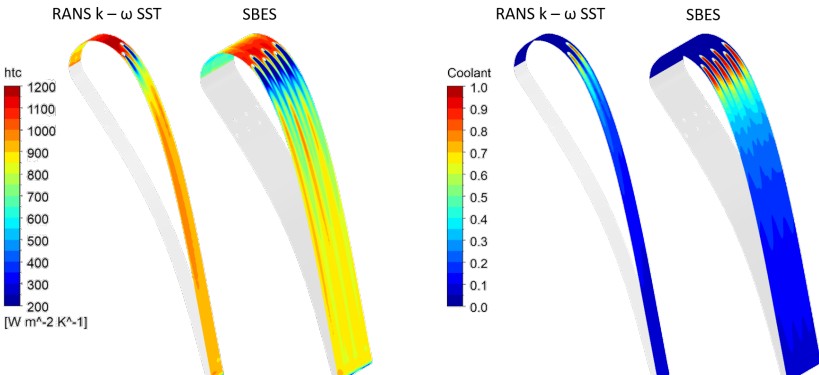

**Figure 25.** Heat transfer coefficient (**left**) and coolant mass fraction (**right**) contours for SS injection case for the different simulation approaches: steady-state fully turbulent $k - \omega$ SST RANS versus unsteady scale-resolving SBES modeling.

## 5. Conclusions

The research presented in this paper focuses on the flow in the highly loaded, film-cooled LS-94 turbine guide vane, designed by the von Karman Institute. First, the RANS fully turbulent $k - \omega$ SST model was benchmarked against transition-sensitive simulations using the $\gamma - Re_\theta$, $\gamma$, and $k - kl - \omega$ transition models with and without film cooling. Then, the investigation moved toward investigating the impact of using an SBES hybrid RANS–LES modeling approach to predict the aerothermal behavior of this kind of turbine vane compared to a more classical RANS setup, focusing only on the cooled vane cases. Comparisons between the simulations and experiments indicated the following:

- The $k - \omega$ SST model tends to overestimate the heat transfer coefficient along the entire blade profile. However, employing a model that tends to overestimate the heat transfer coefficient can be advantageous for ensuring that the design incorporates sufficient cooling and structural robustness to handle higher thermal loads than might occur in operation.
- Caution must be exercised regarding the mixed performance of the transition models in different vane areas. While accurately capturing the behavior at the leading edge, their performance is less consistent elsewhere: the $\gamma - Re_\theta$ and $\gamma$ models significantly underestimate the heat transfer coefficient on the PS and overestimate it on the SS independently of the film-cooling injection, suggesting an excessive turbulence production downstream of the transition point. Further, the SS shows a transition from laminar to turbulent flow. In contrast, the PS exhibits a persistent laminar flow, leading to lower heat transfer coefficient prediction. This is not completely unexpected, as the cooling holes act as a 'trip' in the experiments but not for the transition models, and none of these models are calibrated under 'tripped' conditions. The $k - kl - \omega$ model delays the transition on the SS but is the only transition model capable of reproducing this effect on the PS.

- Careful considerations are necessary concerning the use of the SBES approach in this test case. The SBES results do not show a general improvement in predicting the thermal behavior of the vane, even though an LES treatment of the mainstream would be expected to improve the prediction of mixing-related phenomena, such as film-cooling injection and its impact on the thermal loads. Despite this, it is worth considering that the shielding function might limit the benefits by enforcing a RANS solution in the boundary layer. In addition, this specific test case is highly dependent on turbulence, which for SBES, is generated at the inlet differently compared to RANS.
- Based on considerations of the predictive accuracy over the mesh and computational cost requirements, this study notes that for this highly boundary condition-dependent test case, a low-order analysis, such as the one with the $k - \omega$ SST RANS approach, can be preferred over a high-order one since it produces results similar to SBES and is more conservative than what can be achieved with transition models, which is valuable in the design phase.

In future work, it would be interesting to expand the experimental database to also include film-cooling measurements to better assess the film coverage and discern whether the heat transfer coefficient is determined by the aero flow field or by the film cooling. Concerning the numerical approach, and bearing in mind the associated computational effort, we suggest investigating other operating conditions to assess the sensitivity and robustness of the models to the Reynolds number and the turbulence intensity. In addition, we recommend further investigations to assess the impact of the vertical span of the domain and the turbulence generator on the solution. Furthermore, performing Large-Eddy Simulations to eliminate the shielding function that maintains the boundary layer in a RANS modeling condition, rather than properly resolving turbulence, is also advised.

**Author Contributions:** Conceptualization, S.S. and L.M.; methodology, S.S. and L.M.; formal analysis, S.S. and L.M.; investigation, S.S. and L.M.; resources, R.D.S.; data curation, S.S. and L.M.; writing—original draft preparation, S.S. and L.M.; writing—review and editing, S.S., L.M., R.D.S. and F.F.; visualization, S.S. and L.M.; supervision, R.D.S. and F.F.; project administration, R.D.S. All authors have read and agreed to the published version of the manuscript.

**Funding:** This research received no external funding.

**Data Availability Statement:** The raw data supporting the conclusions of this article will be made available by the authors on request.

**Acknowledgments:** The authors would like to thank their colleagues Lorenzo Palanti and Simone Giaccherini for the valuable discussions about turbulence generation at the inlet of scale-resolving simulations.

**Conflicts of Interest:** Authors Simone Sandrin, Lorenzo Mazzei and Riccardo Da Soghe were employed by the Ergon Research S.r.l. The remaining author declares that the research was conducted in the absence of any commercial or financial relationships that could be construed as a potential conflict of interest.

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
