# Peer review of "Computational Fluid Dynamics Prediction of External Thermal Loads on Film-Cooled Gas Turbine Vanes: A Validation of Reynolds-Averaged Navier–Stokes Transition Models and Scale-Resolving Simulations for the VKI LS-94 Test Case"

_fluids, doi:10.3390/fluids9040091_

Round 1

Reviewer 1 Report

Comments and Suggestions for Authors

CFD Prediction of External Thermal Loads on Film-Cooled Gas Turbine Vanes: A Validation of RANS Transition Models and Scale-Resolving Simulations on the VKI LS-94 Test Case

The authors investigate the heat transfer coefficient for a film cooled turbine blade using CFD. The study uses as a reference experimental data from the VKI under different flow conditions. The authors evaluate a fully turbulent RANS model against different RANS+transition models as well as a hybrid RANS-LES model. The authors observe a relatively large variation in results, with no model being able to capture all main effects reliably.

The k-kl model shows a large error on the prediction of the transition onset even for the un-cooled blade. This is a result of the unexpected strong decay of freestream turbulence upstream of the blade for that model.

The gamma and gamma-Re_theta models predict a reasonable transition location for the un-cooled blade on the suction side but seem to miss transition on the pressure side. This is likely because the cooling holes act as a ‘trip’ in the experiments but not for the transition models. This seems not unexpected, as none of these models appears to be calibrated under ‘tripped’ conditions.

The differences in HTC downstream of transition for the un-cooled case between the models are consistent. The fully-turbulent simulation will result in a earlier/stronger growth of the boundary layer and therefore a thicker boundary layer downstream compared to the transitional simulations. Thinner boundary layers have a higher wall shear-stress and therefore also a higher heat transfer. In other words, the the later transition, the higher the HTC. It is not clear however, why the results even from the SST are above the exp. data.

The RANS study is carried out consistently and demonstrates the capabilities (or lack thereof) of these models.

The motivation and conclusions for the hybrid RANS-LES study (SBES) is less clear. Since transition is one of the main effects in these flows, one would expect that SBES would also need a transition model to capture the upstream laminar part, as the model will operate in RANS mode there. However, even when ignoring this aspect, one would still need to answer the question as to the mode in which the hybrid model is expected to operate. Is the expectation that the model operates in RANS mode upstream of injection and in LES mode downstream? If this is the goal, the authors would need to provide a LES -type mesh resolution in the region downstream of the colling holes.

Figure 21 shows a relatively active LES contribution inside and near the injection hole, but it is not clear what happens further downstream. It looks like the resolved eddies are damped and the model reverts back to RANS? What is the expected benefit from such a set-up compared to a fully turbulent RANS simulation?

From the current reviewer’s perspective, this test case is not suited for a hybrid RANS-LES model like DES/SBES etc. as the main effect of transition cannot be captured. In addition, the mesh resolution provided is most likely/certainly not sufficient to maintain the model in LES mode past the interaction zone with the injection flows. For the SS-injection case, where the SST model gives results very close to the exp. data, it is also not clear what additional benefit the hybrid model is expected to provide. The current reviewer does not see the added value of the hybrid RANS-LES simulation to the publication. It is more of a distraction/inconsistency than a helpful extension of RANS capabilities. It should therefore be removed from the article,

For the current case and with the increasing availability of cheap computing power (GPU), it would be much more interesting to carry out a LES (maybe with a wall model) to see which resolution would be required to improve results compared to RANS – including the transition zone.

Author Response

See the attached rebuttal file.

Reviewer 2 Report

Comments and Suggestions for Authors

The paper deals with a topic which is extremely relevant for the turbomachinery community. The research, which is quite extensive and articulated, is carried out with scientific correctness. The results are interesting and the conclusions are supported by the findings. It is definitely a very good paper and this reviewer recommends it for publication in the Fluids journal. 

Few minor issues should be addressed in order to improve the already high quality of the work:

- Page 2, "Introduction section". It is stated that: "Natural transition: the most common type, due to the inherently unstable nature of the boundary layer at increasing Reynolds number". This is not true, especially in turbomachinery flows. Natural transition is probably the most common transition mechanism in external aerodynamics, that is characterized by high Reynolds numbers and very low free-stream turbulence. In turbomachinery bladings, boundary layers are exposed to relevant free-stream turbulence levels, and transition in attached flow is likely to be triggered by Klebanoff streaks rather than natural instabilities. Thus, it generally occurs in the by-pass mode provided that the boundary layer is attached and not separated. Please rephrase the mentioned paragraph according to these observations.

- Page 8, "Computational methodology section": The authors reports a value of 7.5 mm for the turbulent length scale. Is it the real turbulent length scale (e.g. l=sqrt(k)/(omega)) or is it the integral length scale? Please specify.

  •  
  •  

Author Response

See the attached rebuttal file.

Reviewer 3 Report

Comments and Suggestions for Authors

1)     Literature review is not sufficient, please add more new citations.

2)     The definitions of some parameters in Table 1 are similar to many commonly used parameters. To avoid confusion, it is recommended to use other symbols or add subscripts. For example, pitch (g) is similar to the commonly used symbol for gravitational acceleration, vane height (H) is usually for the head of pumps, fans and compressors.

3)     The mathematical accuracy of different parameters in the article varies, please unify them.

4)     In Figure 4, I feel unreasonable about the given boundary conditions. For example, the GGI indicated in the figure should be periodicity, and the periodicity in the figure should be symmetry? Is what I said correct?

5)     Please add control equations for different turbulence models.

6)     Both 'mesh' and 'grid' are used in the text, please standardize them.

7)     I didn't see your explanation of yplus (y+), it's important and needs to be supplemented.

8)     The difference between experimental values and numerical simulation values has far exceeded the difference between different mathematical models. Does this phenomenon indicate that this study is meaningless? (see Fig. 8)

9)     Pay attention to the issue of italicizing variable fonts.

10)  This study has unique characteristics in analyzing the leakage flow, but further elaboration is needed in terms of flow mechanism.

11)  The conclusion section is not concise enough to serve as a summary. Please make significant revisions. It would be best if several important conclusions could be described separately.

Comments on the Quality of English Language

Acceptable.

Author Response

See the attached rebuttal file.

Round 2

Reviewer 1 Report

Comments and Suggestions for Authors

CFD Prediction of External Thermal Loads on Film-Cooled Gas Turbine Vanes: A Validation of RANS Transition Models and Scale-Resolving Simulations on the VKI LS-94 Test Case

The authors do not consider the question from this reviewer as to the reasoning of using the SBES model in this validation:

·       *  One of the main aspects of this test case is the location of transition. SBES has no provision for transition and therefore behaves much like the underlying RANS model. Why would the authors expect a benefit from using SBES, if the upstream transitional flow is not captured well?

·       *  The question needs to be answered in which regions the SBES model is expected to operate in which mode (RANS/LES)? In case the LES mode is desired say downstream of the injection holes – what would the resolution requirements for LES be and are they satisfied by the current mesh? The authors only show Figure 9 – the LES resolution depicted there is essentially 1 outside the boundary layer which is not relevant. It does not provide any information as to the status of the resolution inside the boundary layer. The shielding function shows a small region of ‘blue’ just downstream of the injection hole but then seems to switch back quickly to RANS. It is also confirmed by Fig. 21 that the resolved eddies die out quickly. When discussing LES resolution of boundary layers, the relevant quantity is the max cell diagonal inside the boundary layer relative to the boundary layer thickness. One would need an absolute minimum of h_max<0.2*delta_BL to maintain LES (likely twice that).

To this reviewer, the application of SBES for this case without a transition model and without a proper mesh design is not suitable. This part of the study should be removed from the article as it does not allow to draw any conclusions.

Author Response

We agree that the location of the transition is one of the key aspects of this test case. Transition can be triggered either by the presence of film cooling (in case of FC injection) or by the presence of the film cooling holes themselves (in case of no FC injection), which disturbs the BL.

The rationale behind the test with SBES was to assess if a high-fidelity approach was able to improve the quality of the prediction (due to the turbulence resolution in the mainstream). The answer seems to be negative, reasonably because the results are driven by the near-wall region rather than the mainstream (as opposed to what happens in gas turbine combustors). The only option we see as effective in improving the results would be LES, which is however unaffordable for the industry, due to both requirements in terms of space discretization close to the walls and time discretization associated with the very small time scales of the eddies in the BL (expected x100 computational effort, while SBES is less demanding).

Concerning the second point, we don't know how confident the reviewer is with SBES (which in the end can be considered an evolution of DES). Contrary to Zonal LES models, here the turbulence resolution is not activated in a specific zone, but rather by the distance from the wall, provided that the mesh is fine enough to resolve the eddies. In other words, the activation of LES is not a matter of mesh sizing, because downstream of the FC holes, the shielding function will force anyway the solution to be RANS in the proximity of the walls. Getting rid of the RANS modelling close to the wall would indeed require a mesh appropriate for LES, but that would lead to the same cost as an LES simulation, which again is unaffordable.

The value of the SBES simulation is that it shows a difference in the HTC, which is associated with the prediction of the coolant coverage. This, to the authors, suggests the necessity to carry out also an experimental investigation dedicated to film cooling estimation.

Best regards,

The Authors

Reviewer 3 Report

Comments and Suggestions for Authors

Can be accepted.

Comments on the Quality of English Language

Can be accepted.

Author Response

Thanks for the effort you put into improving our paper.

The authors